# Effect of Thickness of Ti Coating Deposited by Vacuum Arc Melting on Fatigue Behavior of Aluminum Alloy Al–5%Si

Dmitrii Zaguliaev [1,*], Yurii Ivanov [2], Suresh Gudala [1], Oleg Tolkachev [2], Krestina Aksenova [1], Sergey Konovalov [1] and Vitaly Shlyarov [1]

1 Siberian State Industrial University, Novokuznetsk 654007, Russia; gsuresham@gmail.com (S.G.); 19krestik91@mail.ru (K.A.); konovserg@gmail.com (S.K.); shlyarov@mail.ru (V.S.)
2 Institute of High Current Electronics SB RAS, Plasma Emission Electronics Laboratory, Tomsk 634055, Russia; yufi55@mail.ru (Y.I.); tolkachev@tpu.ru (O.T.)
* Correspondence: zagulyaev_dv@physics.sibsiu.ru; Tel.: +7-(913)-421-28-88

**Abstract:** Fatigue strength tests of Ti-coated aluminum alloys with a thickness of 1 μm, 3 μm, and 5 μm were conducted to investigate the effect of the coating thickness on fatigue strength. Under the same applied stress amplitude, the optimum thickness with the most-extended fatigue life was around the coating thickness of 5 μm. This may be attributed to the good resistance to surface cracks under repeated loads. The results suggested that a lower fatigue life of a coating thickness of 1 μm results from the fracture of the coating layer under the strong influence of the deformation of the substrate. This could be due to the higher tensile residual stress induced in the substrate near the coating layer and substrate interface. The titanium coating restricted the initiation of offsets and cracks beneath the surface of the specimen, which may be attributed to the high strength of the Al–5%Si substrate, good flexibility, and strong adhesion, which provided sufficient compressive stress to suppress slip band protrusions. The fatigue life and fatigue limit increased proportionally to the thickness of the titanium coating due to changes in the surface roughness and adhesion capability.

**Keywords:** titanium coating; aluminum alloy; Al–5%Si; microhardness; fatigue life; failure analysis





## 1. Introduction

Aluminum alloys are widely used in aerospace, railways, automobiles, and other fields due to their high strength, low density, and low specific gravity. Al alloys have lower fatigue endurance than steel. In recent years, researchers have developed a variety of surface treatments, such as coating [1], nitriding [2], ion implantation [3], shot peening [4], etc., to improve the compressive residual stress on the surface to prevent crack initiation and growth. Coating process research focuses on environmentally friendly technologies that provide corrosion and wear protection to ensure their competitiveness and discover new potential applications. However, it is critical to investigate the impact of new coatings on the fatigue resistance of the base material when the component is subjected to cyclic loading [5–9]. Surface modification of aluminum alloys is widely used in the aerospace industry in order to improve mechanical and fatigue performance. The addition of titanium to Al alloys used in aerospace components increases surface hardness and can also be effective against corrosion.

In many cases, micro-arc oxidation treatments improve aluminum alloys' corrosion and wear resistance [10–13], but invariably reduce the fatigue property. The microcracks, pores, and notches on the coating are prone to fatigue cracks, lowering the fatigue property of aluminum alloys [14]. Usually, fatigue cracks form near the coating–substrate interface due to tensile substrate residual stress (SRS), which eventually reduces the fatigue strength [15]. Researchers have studied extending the fatigue life of parts and structures in recent decades to lessen the material degradation. The surface engineering method is an efficient way to combat the adverse effects of fatigue loads [16]. The parameters influencing

fatigue life include contact pressure, surface hardness, axial stress frequency, humidity, frictional force, etc.; contact materials are the most important [17–21]. Puchi-Cabrera et al. [22] investigated the effect of TiN coating on the mechanical properties of 316L stainless steel. The authors reported a 60 MPa improvement in the fatigue limit with no significant changes in the yield and tensile strength. The increase in the fatigue strength was attributed to compressive residual stresses within the coating. Ryabchikov et al. [23] revealed that the $Al_3Ti$ intermetallic layer on the aluminum with a 5 μm thickness improved the mechanical properties and wear resistance.

A few authors [24,25] have reported on the corrosion and mechanical performance of coatings developed using the cold spraying method. Wang et al. investigated the corrosion characteristics as the coating thickness varied. Increasing the coating density has been shown to decrease the corrosion current [25]. Due to the higher silicon amount, Al-Si alloys have better tribological properties than other aluminum alloys [26,27]. As a result of the significant anisotropy and modest string energy between Al and Si, the lamella structure in the Al-Si eutectic was tied [28]. It has been shown that increasing the Si concentration reduces thermal expansion and can be employed as a hard phase for improving the alloy's wear resistance [29]. Furthermore, some research studies have investigated the importance of various phases on mechanical and corrosion properties [30,31].

Similarly, another study by Figueroa et al. [32] investigated that nitrogen ion implantation results in a modified layer of aluminum nitride on the material surface. This leads to the better wear resistance of AA7075-T73 alloy under lubrication conditions while lowering the friction coefficient and wear rate by 33%. Many studies investigated aluminum alloys by depositing the TiN, ZrN, WC/C, and NieP coatings by physical vapor deposition (PVD) to improve fatigue strength [33–37].

In earlier studies, it was observed that pure titanium on aluminum alloy substrates provided better surface properties. A coating of pure titanium would be a better alternative to increase the corrosion and fatigue resistance of aluminum alloys. Furthermore, titanium is widely used for its corrosion prevention, fatigue resistance, weight savings, replacement costs, and life cycle cost benefits. The effect of the coating thickness on crack initiation and crack growth is unclear in the literature. Because crack initiation has the most-significant impact on fatigue life, extensive investigation is needed. The main goal of this research was to investigate the microstructure of the coatings and to study the effects of the titanium coating thickness on the mechanical characteristics. Furthermore, the fatigue life and fracture morphologies were investigated with the increased coating thickness.

## 2. Materials and Methods

In the present study, Al–5%Si aluminum alloy of chemical composition wt%: Al balance, Si 4.0–6.0%, 1.5–3.5 Cu%, <1.5 Zn%, <1.0 Fe%, <0.5 Ni%, 0.2–0.8 Mn%, 0.05–0.20 Ti%, and 0.2–0.8 Mg%, was used as the substrate material. The deposition of a pure titanium coating with a thickness of 1 μm, 3 μm, and 5 μm on the surface was carried out by vacuum arc plasma-assisted deposition (KNVINTA Installation). In the installation, a cathode made of commercially pure titanium grade VT1-0 was used for depositing pure titanium films with an electromagnetic drop fraction filter. The coated samples were cut for fatigue tests according to the ASTM E8/E8m-16a standard [38]. The fatigue tests of the Al–5%Si samples were carried out according to the scheme of asymmetric cantilever bending on a setup developed at SibSIU [38]. The preliminary tests were conducted to quantify the mechanical fatigue inherent in the fatigue machine and tooling according to ASTM E 739 [39]. In the present study, the maximum stress applied on the specimens during cycling was lower than the limit of elasticity (85%) estimated from preliminary tests. The average value of the fatigue strength was estimated based on three tests. The objective was to maintain the maximal stress value below the elastic limit while remaining within the limited fatigue area to achieve specimen failure between the estimated cycles. Later, the fractography analysis of the broken samples was performed using a scanning electron microscope (SEM) and transmission electron microscope (TEM). To assess the coating thickness and microstruc-

tural studies, a few samples were polished with the alumina suspension after being ground with 400, 800, 1200, and 2000 grit emery papers.

The Al–5%Si substrate samples were etched with chemical agents to know the surface characteristics of the Al–5%Si aluminum alloy samples. The etchant was prepared and applied with 5 mL of nitric acid, 3 mL of hydrochloric acid, 2 mL of hydrofluoric acid, and 190 mL of distilled water in order to reveal the microstructural details. The porosity of the coatings was measured with an optical microscope with the attached image analyzer. The surface morphology, cross-section structure, thickness, and fatigue failure of the developed coatings were characterized with a scanning electron microscope (SEM) equipped with energy-dispersive X-ray analysis (EDX) (Philips-SEM-515, EDAX ECON IV, 150 kV, Bologna, Italy) and transmission electron microscopy (JEM-2100F instrument, JEOL, Akishima, Japan). Furthermore, the coating was characterized for the possible formation of spikes, droplets, pores, and microcracks. The porosity of the coatings was analyzed using an optical microscope attached to a biovis image analyzer (ARTRAY, AT 130, Tokyo, Japan).

### 3. Results and Discussion

According to the metallographic studies presented in [39–41], selective etching methods can be used for the Al-Si alloy system. $Al_5SiFe$ phases are like lamellar inclusions; the inclusions of $\alpha(Al_{15}(FeMn)_5Si)$ are in a regular polyhedron shape; iron particles are formed in the shape of Chinese characters; silicon particles are in the shape of oval inclusions. It was found that, in the Al–5%Si sample, a large number of inclusions (dashed circles) were predominantly present, which were in a lamellar shape (Figure 1). Figure 1 shows various sizes of the pores, droplet formation, and spikes. Droplets formed as microparticles, and a few spikes were clearly visible on the coating surface. Pores may have formed when solidified droplets broke off after deposition was complete. The formed pores were estimated to be in the 0.05–10 μm-diameter range.

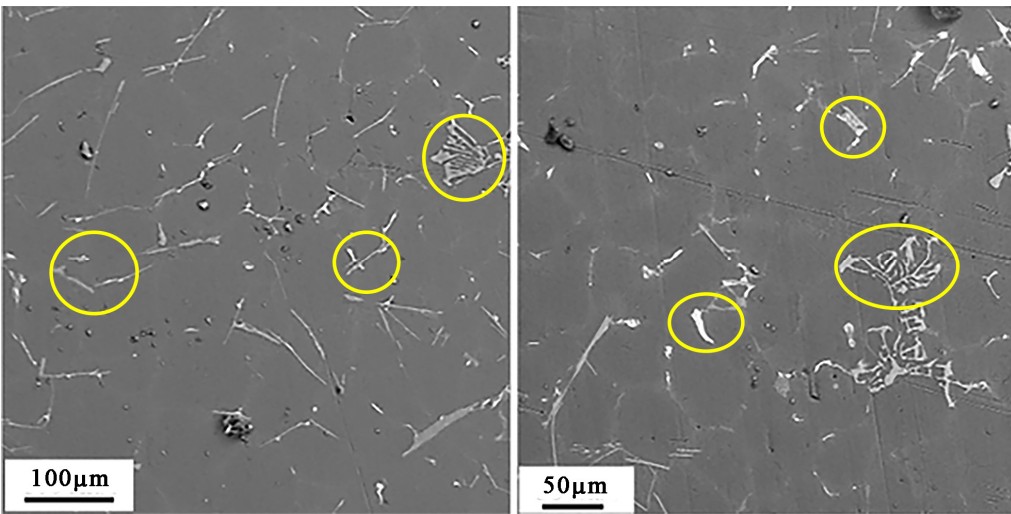

**Figure 1.** SEM image of etched Al–5%Si sample.

The titanium-coated sample is shown in Figure 2a. A contact nanoprofilometer was used to measure the surface roughness of the substrate and coated samples. The surface roughness of the Al-Si substrate alloy before the deposition of a titanium layer was estimated to be Rz = 1.21 μm; Ra = 0.147 μm. The surface roughness of a Ti coating of 1 μm was Rz = 0.660 μm; Ra = 0.099 μm. The roughness decreased as the titanium film's thickness increased from 1 μm to 3 μm (Rz = 0.452 μm; Ra = 0.071 μm). The surface roughness was further reduced as the coating thickness increased to 5 μm (Rz = 0.360 μm; Ra = 0.056 μm). The lower surface roughness made it difficult to form stress concentrators, which contributed to the initiation of microcracks during the fatigue tests.

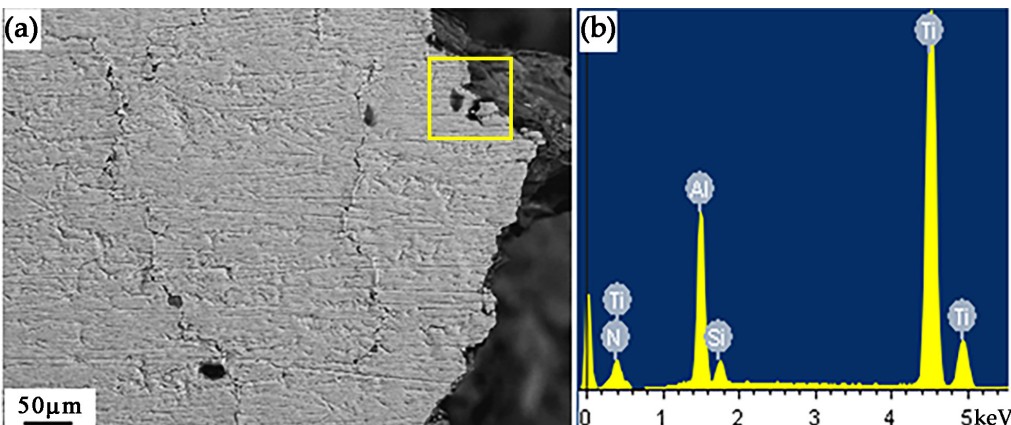

**Figure 2.** (**a**) SEM image of Ti-coated Al–5%Si sample of 1 μm thick. (**b**) EDS spectra of the area marked in (**a**).

To know the elemental composition of the coating sample, an EDS analysis was carried out. As shown in Figure 2b, titanium and aluminum were predominantly present in the selected area. As observed from Figure 3a, numerous microcracks were observed on the surface after the fatigue testing. However, with the increase in the thickness of the titanium coating, the surface roughness was found to decrease, and microcracks were barely visible. To assess the microcracks of all fatigue samples tested, the SEM images were analyzed with the crack width calculation. It was found that the open crack width of the Al–5%Si sample was 12.5 μm (Figure 4a). The width of the open crack of the sample with a 1 μm-thick coating was 2.5 μm. The width of the open crack of the sample with a 3 μm-thick coating was 2 μm. Similarly, the width of open crack of the sample with a 5 μm-thick coating was 1.25 μm. The above results indicated that the deposited coating showed better bonding with the substrate. Even after the fatigue tests, the coatings remained bonded to the substrate. However, with the increase in coating thickness, the resistance to crack initiation and crack growth increased substantially. Thereby, the fatigue life of the sample increased. In this study, the failure of the Al–5%Si sample occurred at $1.14 \times 10^5$. However, the sample with a 5 μm-thick coating of titanium sustained a higher number of cycles ($2.45 \times 10^5$).

The coating was more likely to crack under fatigue load when the applied load amplitude was large, which lowered the capacity of the coating to resist oxidation. Meanwhile, the fractured coating increased the stress concentration and shortened the fatigue life of the substrate. However, the coating had almost no impact on the fatigue life of the substrate when the applied load amplitude was minimal. The defects that accumulated within the substrates during the fatigue test caused slip bands to form in the crystalline substrate. When the slip bands propagated to the film/substrate interface, they formed a step-like protrusion on the film's surface, which eventually formed a crack. During the fatigue test, the defects within the substrates led to the formation of slip bands. As the slip bands propagated into the interface, a step-like protrusion was created on the surface of the coating, which eventually formed a crack.

The magnified SEM images of the fatigue-tested samples are shown in Figure 4. The transverse cracks that occurred in all specimens concerning the longitudinal axis were analyzed to evaluate the effect of the fatigue loading. With the increase in the thickness of the coatings, the transverse cracks in all samples were reduced substantially. The average size of the fragments of the Al–5%Si sample and 1 μm Ti-coated sample was 270 μm and 65 μm, respectively (Figure 4a,b). The fragments of the 3 μm Ti-coated sample were in the range of 4.5–18 μm (Figure 4c). Similarly, the size of the fragments in 5 μm Ti-coated sample was in the range of 1–1.5 μm. From the analysis, it is clear that the surface characteristics of the Al–5%Si sample played a vital role in the fatigue behavior.

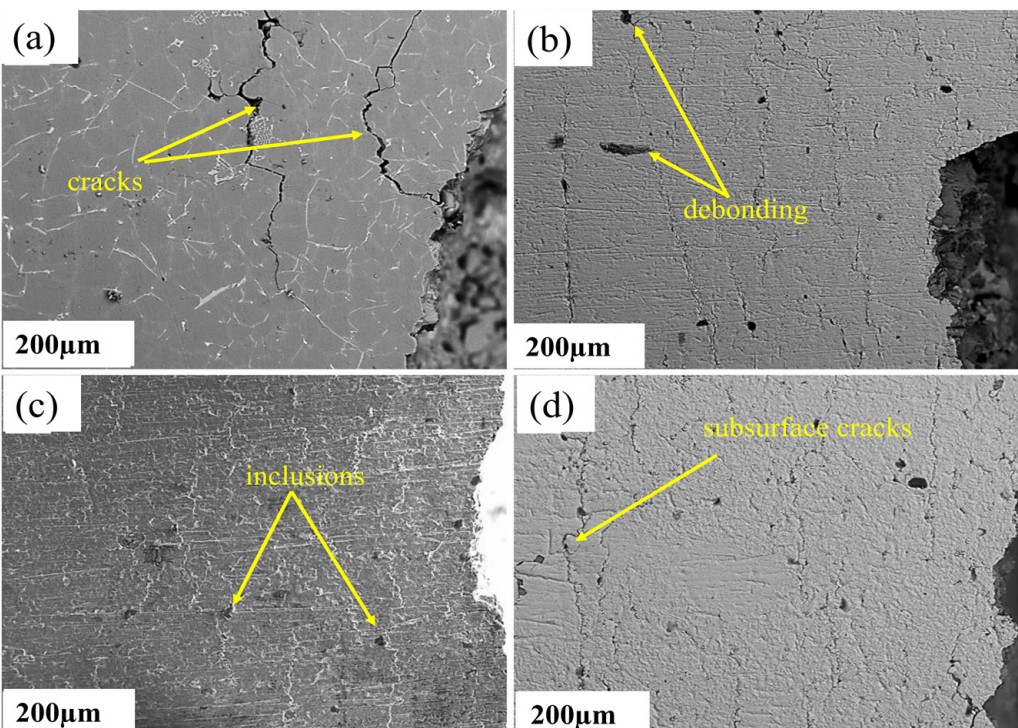

**Figure 3.** SEM images of surfaces showing crack propagations after fatigue failure. (**a**) Al–5%Si sample; (**b**) 1 μm Ti coating; (**c**) 3 μm Ti coating; (**d**) 5 μm Ti coating.

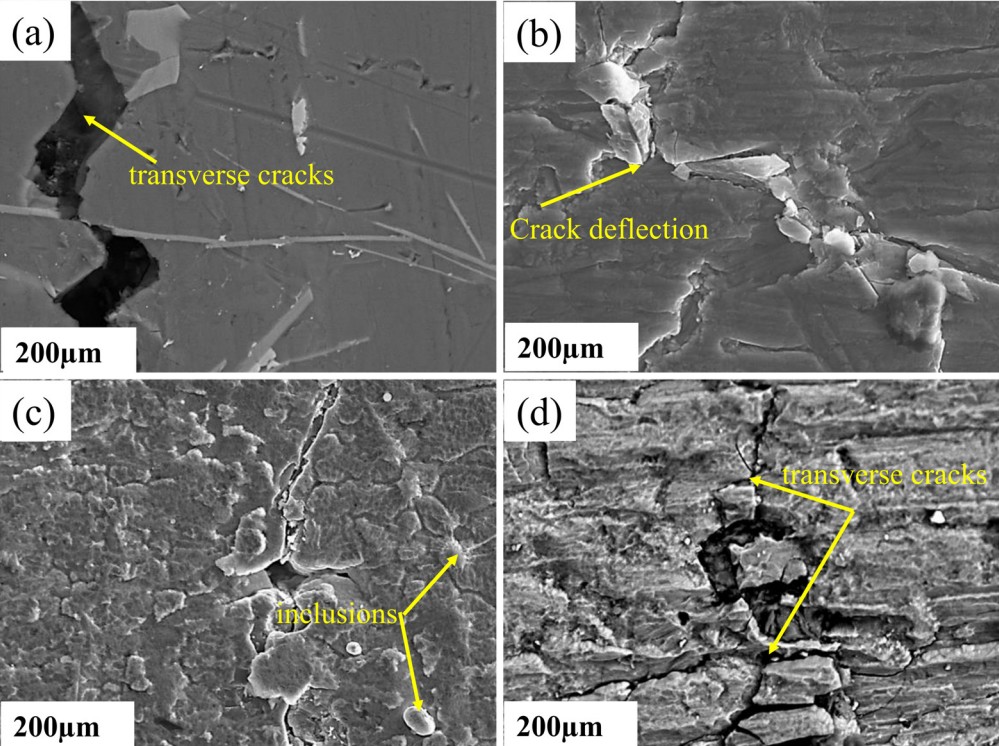

**Figure 4.** Magnified SEM images of surfaces showing microcracks after fatigue failure. (**a**) Al–5%Si sample; (**b**) 1 μm Ti coating; (**c**) 3 μm Ti coating; (**d**) 5 μm Ti coating.

Figure 5a,b depict the fatigue striation morphologies in the crack-growth zone. The crack propagated forward as a fatigue striation in the crack-growth zone. The greater the grain boundary, the more energy the fatigue crack propagation consumed, reducing the

crack growth rate. The fatigue cracks grew slowly as a result of the cyclic vibration load. The fatigue striation spacing approximated the crack growth rate. The smaller the spacing between fatigue striations, the slower the fatigue crack growth was [42]. The fracture zone was mainly made up of dimples, micropores, and tear ridges on the side of the dimples. Many second-phase particles can be seen at the bottom of the dimples in Figure 5a. During the vibration in the repeated loading, some particles were torn into two, and others were separated from the bottom of the dimples. Furthermore, it was observed that the dimples of the uncoated samples were more prominent and deeper than the coated samples, indicating improved fracture toughness.

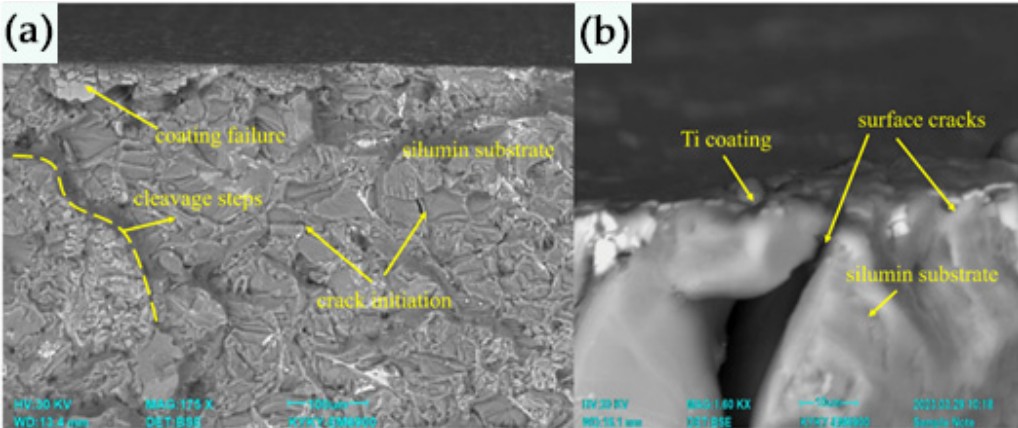

**Figure 5.** (**a**) SEM image of 5 μm Ti-coated fractured sample. (**b**) Crack propagation.

Figure 6a indicates the Ti-coated sample before the fatigue analysis. The crack initiation was transferred from the surface to the subsurface in the fatigue crack initiation, as shown in Figure 6b,c. The microstructures might primarily influence this in the surface layer. With the difference in the elastic moduli and residual stresses between the particles, the inclusions and surrounding second-phase particles caused a local stress concentration during each vibration cycle [43]. As a result, many small voids aggregated into microcracks and formed fatigue cracks in the vicinity of the inclusion and second-phase particles. Transmission electron microscopy was used to examine the coated samples that were subjected to fatigue tests. It was clearly seen that the fatigue tests of the samples were accompanied by a significant transformation of the Al–5%Si layer adjacent to the titanium film (Figure 7). The maximum thickness of the contact layer varied from 0.5 μm to 2.0 μm. It increased monotonically with the increase in the number of fatigue test cycles. The contact layer had many micropores and microcracks, which increased as the number of fatigue test cycles increased.

X-ray microanalysis was performed to investigate the elemental composition of the film and the contact layer (mapping method). Figure 8 shows that aluminum atoms formed the contact layer and contained inclusions enriched with silicon, iron, copper, and zinc atoms. The film's main component was titanium, but it also contained copper and oxygen atoms. As a result, it is reasonable to assume that the Al–5%Si fatigue tests were accompanied by the oxidation of the titanium film deposited on the surface of the samples. The phase composition of the deposited film was examined using diffraction electron microscopy techniques, such as the dark-field technique and the indexing microelectron diffraction patterns (Figure 8).

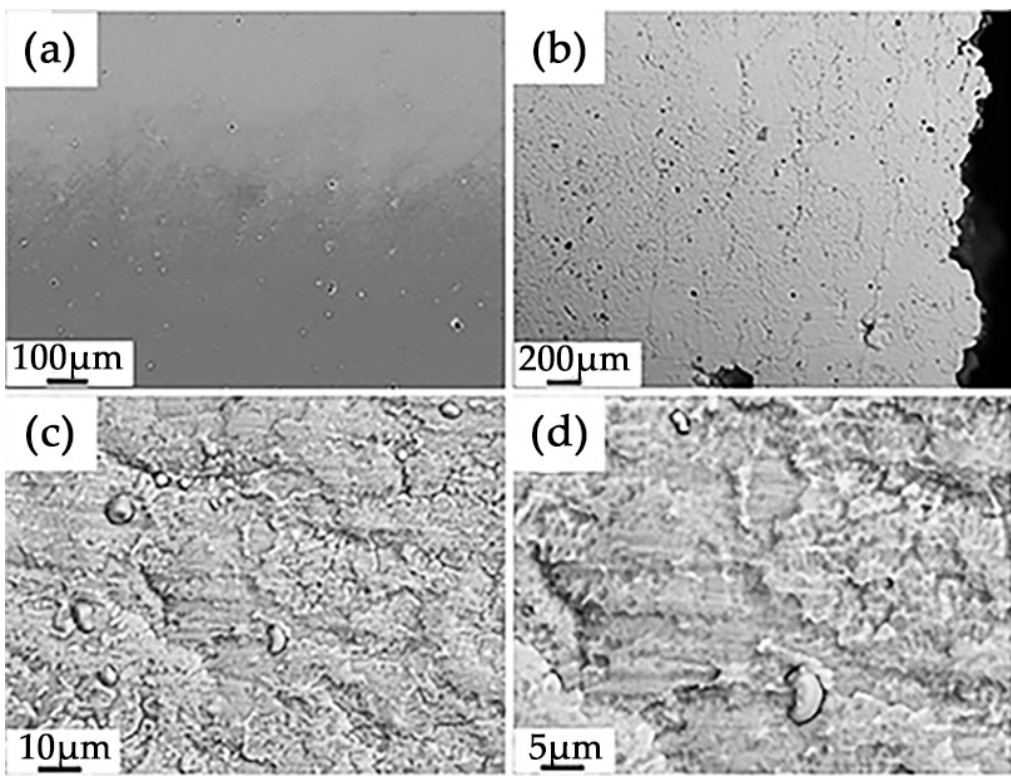

**Figure 6.** SEM images of the surfaces of 5 μm Ti-coated sample. (**a**) Initial state; (**b**–**d**) after fatigue test.

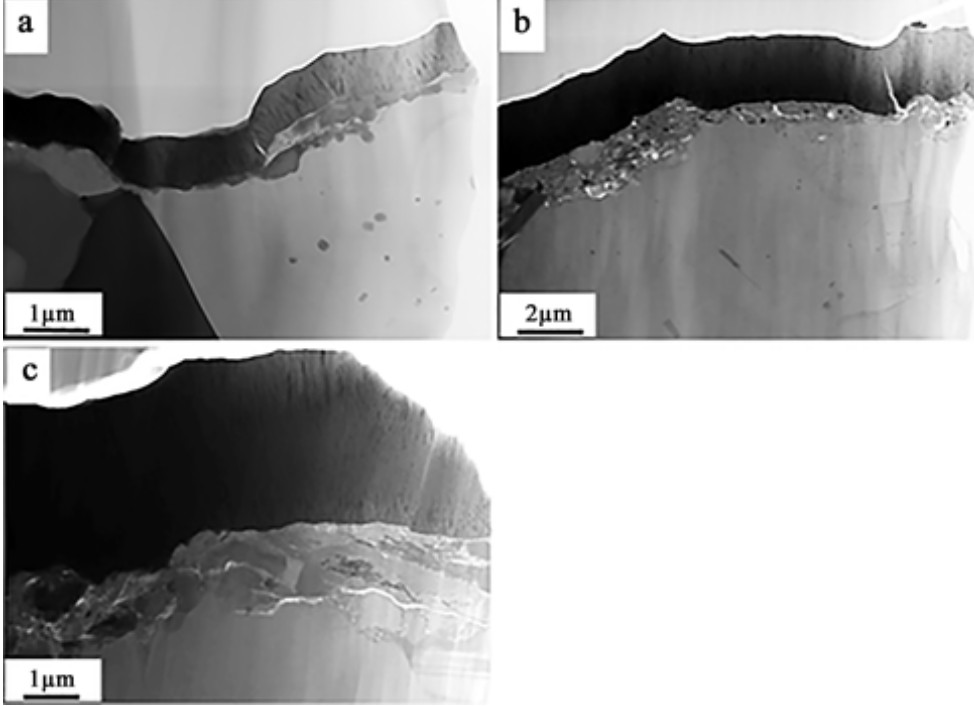

**Figure 7.** Transmission electron microscope images of Ti coatings after fatigue tests: (**a**) 1 μm $(1.1 \times 10^5)$; (**b**) 3 μm $(2.0 \times 10^5)$; (**c**) 5 μm $(2.5 \times 10^5)$.

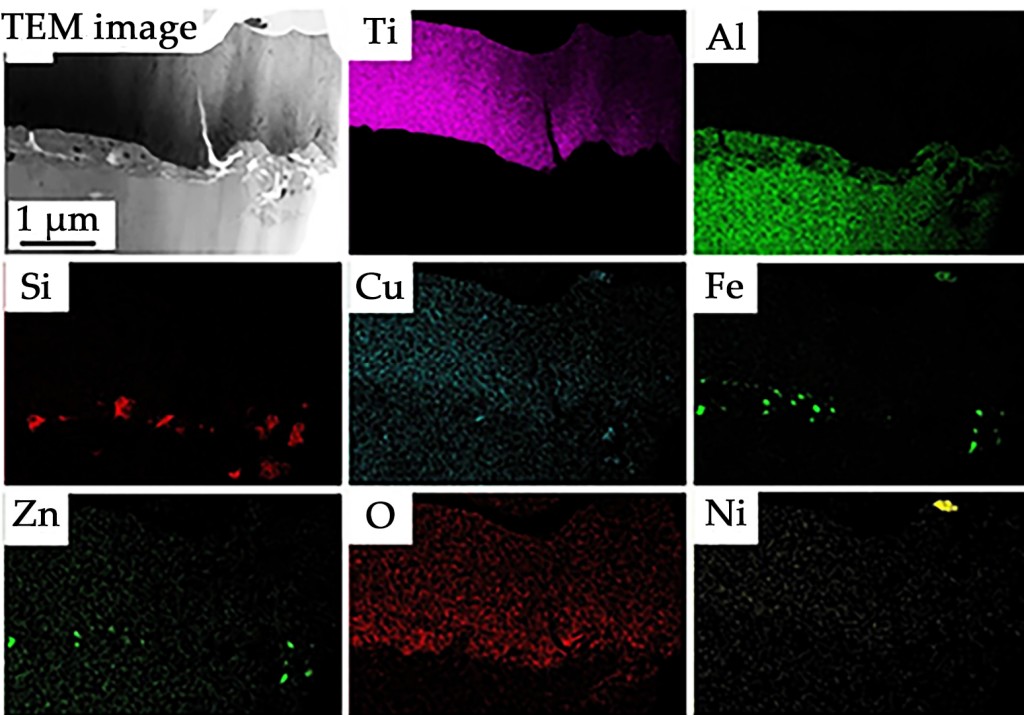

**Figure 8.** Elemental mapping of the 5 μm Ti coating.

The nanoprecipitation was uniformly distributed within the grains. The many dislocations gathered around the nanoprecipitations. The nanoscale precipitations can bind the dislocation movements. As a result of the formed deformation, new deformations were formed to locate the plastic deformation, leading to an increase in dislocation density. It was established that the vacuum arc plasma-assisted deposition film formed on the surface of Al–5%Si had a columnar structure with a column thickness of 20–35 nm (Figure 9a,b). The microelectron diffraction pattern obtained from the film (Figure 9c) showed that the main phase of the film was titanium oxide with the formula TiO. As a result, the Al–5%Si fatigue tests were accompanied by the oxidation of the titanium film deposited on the samples. The fatigue life of the samples was found to increase from 110,784 cycles to 245,274 cycles as the thickness increased from 1 μm to 5 μm. The Al–5%Si samples with no titanium film on the surface lasted fewer cycles (N = 113,742) (Figure 9). As shown in Figure 10, a defective substructure in a titanium-coated Al–5%Si sample subjected to fatigue loading resulted in a significant transformation of the layer adjacent to the fracture surface. The presence or absence of a titanium layer had the most-significant influence on the substructure. In the absence of the titanium coating, the scalar density of the dislocations was $1.5 \times 10^{10}$ cm$^{-2}$, as shown in Figure 10 as a network-type dislocation substructure. The formed dislocations formed tangled structures as they accumulated near the grain boundaries. Furthermore, subgrain boundaries were formed around the dislocation gathering areas. The nanoprecipitations were observed on the surface layer, and some crack deflection and breaching morphologies were attributed to the interactions between the crack tip and the heterogeneous microstructure. The produced structures could improve the fracture toughness by restricting crack propagation and lowering the crack growth rate.

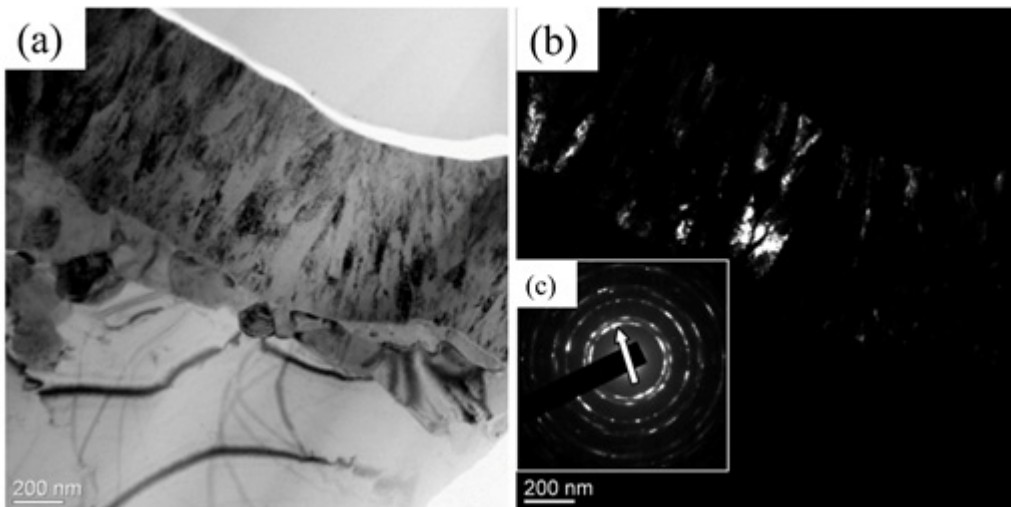

**Figure 9.** (**a**) TEM image of the Ti coating after fatigue tests at $1.1 \times 10^5$. (**b**) Dark field in the [111] TiO reflection. (**c**) Microelectron diffraction pattern (the arrow indicates the reflection in which the dark field was obtained).

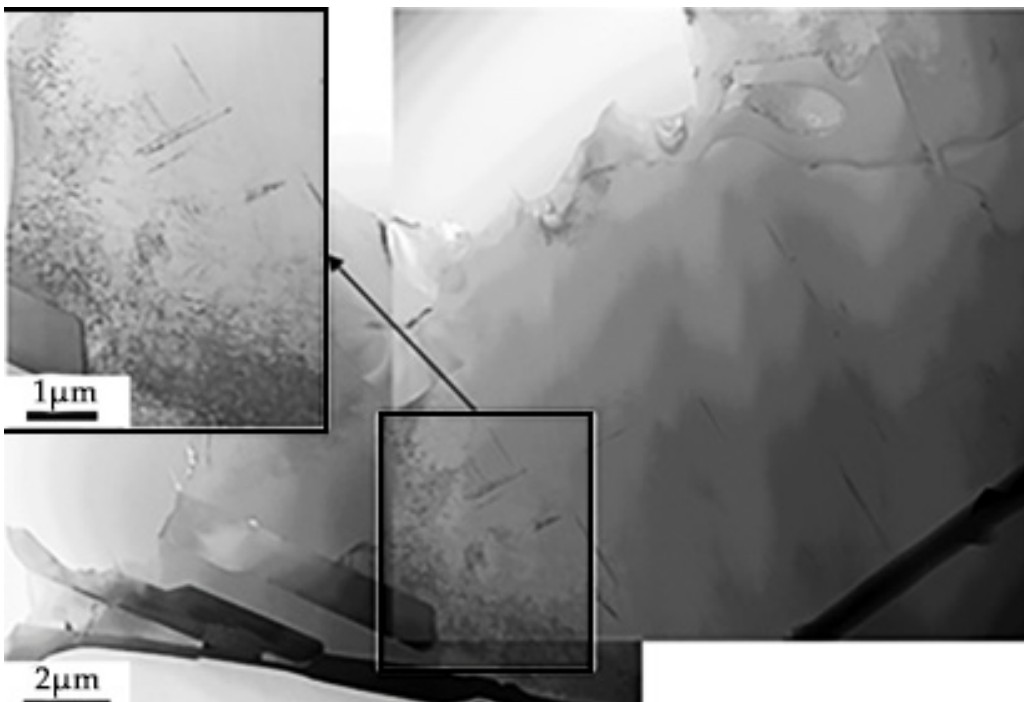

**Figure 10.** TEM image of the fracture surface of the Al–5%Si substrate without the titanium coating under fatigue loading.

The formation of internal stress fields was influenced by the fatigue loading. The bending extinction contours in the TEM image indicated the presence of identified stress fields, as shown in Figure 11a. The aluminum grain interface and second-phase inclusions dominated the thin foil structure. In the volume of the bending extinction contours, both primary aluminum grains and second-phase inclusions were found to be elastically stressed (Figure 11b).

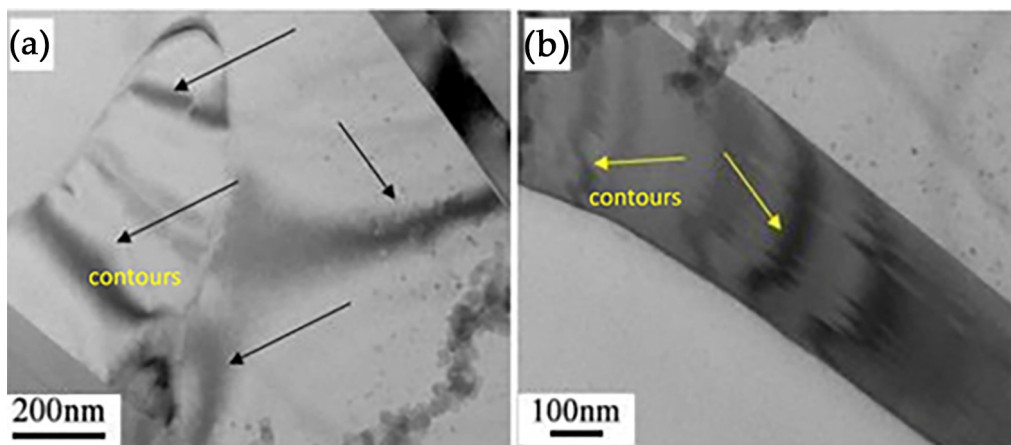

**Figure 11.** TEM image of fractured sample of (**a**) Al–5%Si and (**b**) Al–5%Si with a 5 μm titanium coating.

The formation of a subgrain structure upon Al–5%Si destruction occurred with and without the deposition of a titanium film on the working part of the Al–5%Si samples (Figure 12). It was noted that the Al–5%Si sample with a 1 μm-thick titanium film was destroyed after fewer cycles (N1 = 110,784) than the Al–5%Si sample without a titanium film (N0 = 113,742). This implies that the formation of a subgrain structure in the Al–5%Si destruction zone was caused by the joint deformation of two materials: Al–5%Si and titanium. The thickness of the layer with a subgrain structure varied within (6–13) μm and increased with the number of cycles to destruction (thickness of the deposited titanium film). A layer with a subgrain structure had a gradient structure, which means that subgrains with minimal (0.1–0.2 μm) sizes were located on the fracture surface (Figure 12). The size of the subgrains increased as the distance from the fracture surface increased to (1–1.5) μm. The heterogenous grain structure is an essential factor in improving the fatigue performance. Near the subsurface, fine and coarse grains were observed as heterogenous grain structures. Due to the higher tortuosity, it can be stopped in the fine-grained zone during the initiation of the crack. Furthermore, crack propagation can be reduced with the formation of heterogenous grain structures, which further lowers the potential risk of microcrack initiation.

A layer with a subgrain structure was visible at long (30–40 μm) distances from the Al–5%Si destruction surface. Figure 13a,b depict the formation of subgrains at the interface of two phases: aluminum and silicon. As a result, the formation of subgrains, in this case, resulted from the joint deformation of the two phases with different mechanical properties. Diffraction electron microscopy revealed that the subgrains were separated by boundaries with small-angle misorientation (Figure 14). The azimuthal component of the total misorientation angle of fragments varied within 3–5°, according to the analysis of the microelectron diffraction patterns obtained from such a structure. The defects that accumulated within the substrates during the fatigue test caused the crystalline substrate to form slip bands. When the slip bands spread to the film/substrate interface, they formed a step-like protrusion on the film's surface, which eventually formed a crack.

As previously stated, the fatigue tests caused the formation of elastic stress fields in the material, which were revealed by bending extinction contours in the electron microscopic images of the structure of the thin foils. Figure 15a,b depict the typical images of a subgrain structure with bending extinction contours as the titanium film thickness increased. The variation of the subgrain structure with the film thickness is indicated as arrows.

The presence of prismatic dislocation loops was discovered during the studies of the dislocation substructure in the layer adjacent to the fracture surface of the Al–5%Si samples with a pre-deposited titanium film (Figure 16). The lack of contrast within the loops, which is typical of stacking faults, indicated that the loops were formed by complete dislocations [44,45]. A large number of vacancies or interstitial atoms can combine to form

dislocation loops [46]. Vacancy dislocation loops are formed during the quenching of metal crystals from high temperatures.

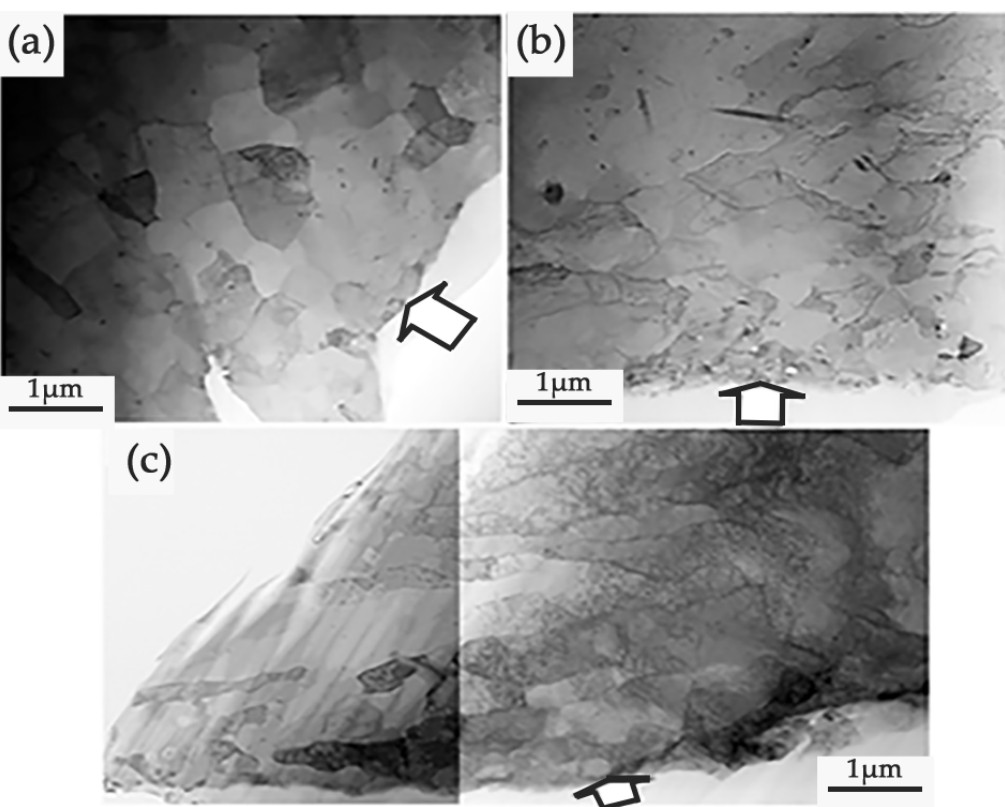

**Figure 12.** Formation of layer on the fracture surface of a Al–5%Si sample with a preliminarily deposited titanium film of thickness (**a**) 1 μm, (**b**) 3 μm, and (**c**) 5 μm.

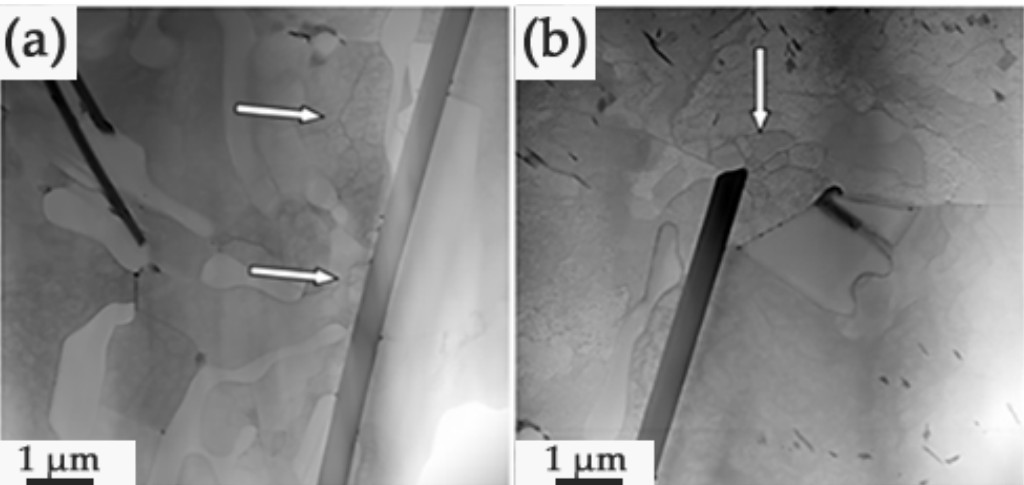

**Figure 13.** The structure of a Al–5%Si sample with a 3 m titanium film at a distance of (**a**) 30 μm and (**b**) 40 μm from the fracture surface.

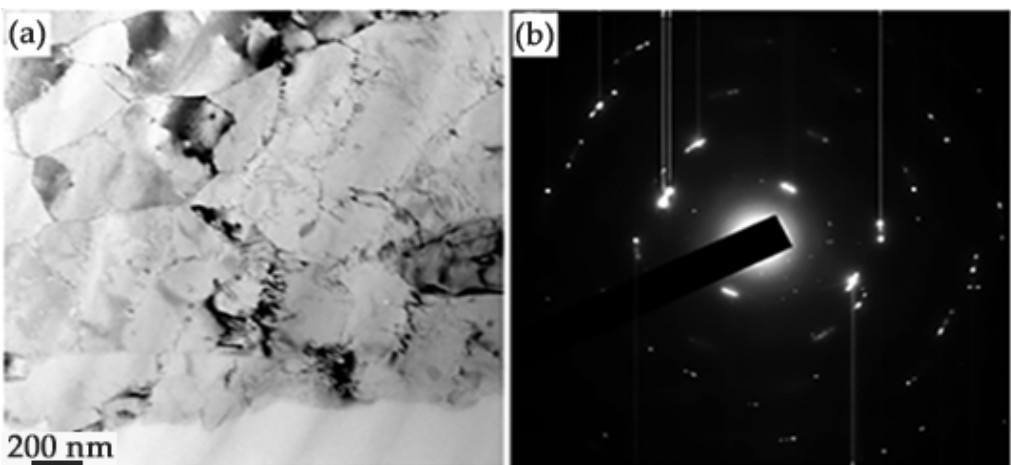

**Figure 14.** TEM images of the subgrain structure formed in the region of the destruction surface of the Al–5%Si sample with a 5 μm thickness: (**a**) bright field image; (**b**) microelectron diffraction pattern on foil section shown in (**a**).

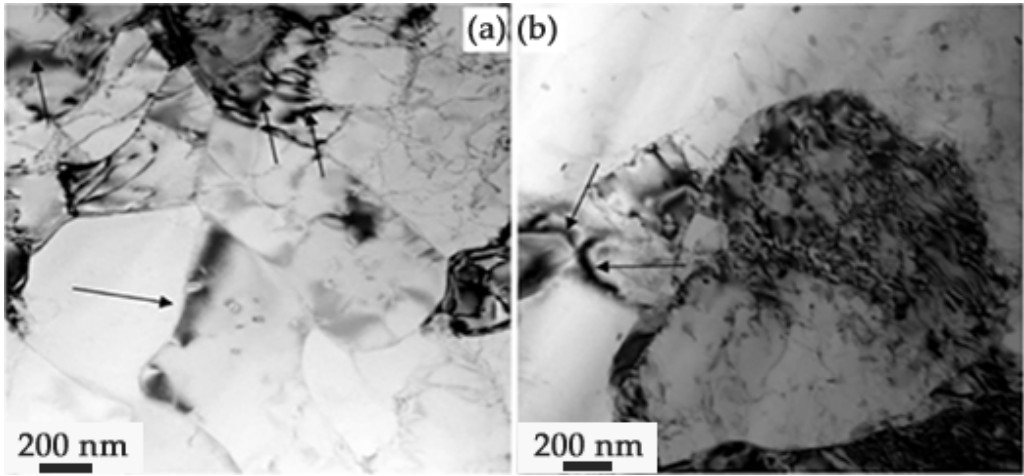

**Figure 15.** Bending extinction contours in the fractured Al–5%Si sample of (**a**) 3 and (**b**) 5 μm titanium film thickness.

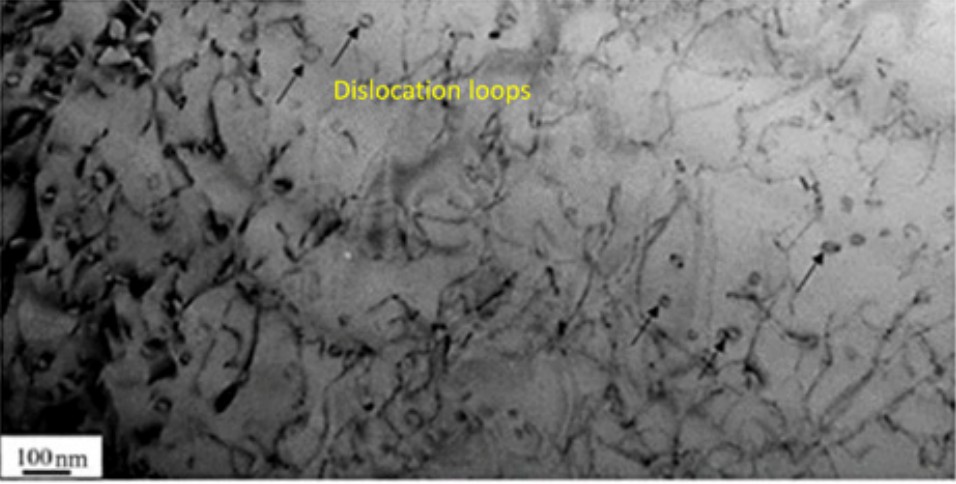

**Figure 16.** Electron microscopic image of the dislocation substructure of 5 μm titanium-coated Al–5%Si sample (black arrows indicate dislocation loops).

## 4. Conclusions

From the investigations carried out with the fatigue analysis of the Ti-coated Al–5%Si substrate, the main conclusions were obtained as follows:

1.  Microcracks formed primarily at the inclusion–matrix interface during the Al–5%Si fatigue tests, and the transverse scales of the opened microcracks fixed on the side surface of the Al–5%Si samples were identified to depend on the thickness of the deposited titanium film.
2.  The open crack width decreased from 2.5 μm to 1.25 μm as the film thickness increased from 1 μm to 5 μm. It was indicated that fatigue tests of the coated samples are accompanied by the forming of a fragmented structure with dimensions ranging from 4.5 μm to 18 μm on the side surface of the samples.
3.  The formation of the fragments of dimensions ranging from 1–1.5 μm was obtained within the volume of the fragments.
4.  With the increase of the coating thickness from 1 μm to 5 μm, the contact layer varied from 0.5 μm to 2.0 μm and further increased monotonically with the number of fatigue test cycles.
5.  In this study, the film formed on the surface of Al–5%Si had a columnar structure with a column thickness of (20–35) nm, indicating that fatigue tests of titanium-coated Al–5%Si were accompanied by the oxidation of a titanium film, which led to the formation of TiO.
6.  The formation of a subgrain structure in the Al–5%Si destruction zone was caused by the joint deformation of two materials: Al–5%Si and titanium. The layer thickness with a subgrain structure varied within 6–13 μm and increased with the number of cycles to destruction.

**Author Contributions:** Conceptualization, D.Z. and Y.I.; methodology, Y.I.; validation, O.T. and K.A.; formal analysis, S.G. and V.S.; investigation, D.Z., Y.I., S.G. and V.S.; resources, D.Z. and S.K.; data curation, Y.I. and O.T.; writing—original draft preparation, Y.I., S.G. and V.S.; writing—review and editing, D.Z. and S.G.; visualization, K.A.; supervision, S.K.; project administration, D.Z. All authors have read and agreed to the published version of the manuscript.

**Funding:** The research was supported by a grant from the Russian Science Foundation No. 19-79-10059, https://rscf.ru/en/project/19-79-10059/ (accessed on 1 September 2023). The research was carried out using the equipment of the Center for Sharing Use "Nanomaterials and Nanotechnologies" of Tomsk Polytechnic University supported by the RF Ministry of Education and Science Project #075-15-2021-710.

**Institutional Review Board Statement:** Not applicable.

**Informed Consent Statement:** Not applicable.

**Data Availability Statement:** Not applicable.

**Conflicts of Interest:** The authors declare no conflict of interest.

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
