# Peer review of "Effect of Thickness of Ti Coating Deposited by Vacuum Arc Melting on Fatigue Behavior of Aluminum Alloy Al–5%Si"

_coatings, doi:10.3390/coatings13101764_

Round 1

Reviewer 1 Report

The paper "Effect of thickness of Ti-coating deposited by vacuum arc melting on fatigue behavior of aluminum alloy Al–5%Si" presents very interesting aspects in terms of microstructure and can be published in Coatings Journal after some minor updates:

1. The introduction is generally well written, but there are some aspects that need to be added: 1) Please add other examples of methods of coatings, like APS (athmospheric plasma spaying), suitable for mechanical testing, and 2) present the influence of Al and Si in terms of microstructure, mechanical, and corrosion properties. Suggested references: 10.4028/www.scientific.net/JERA.37.23; https://doi.org/10.3390/su132111771.

  1. Line 82: Please add more parameters for SEM and EDS analysis.

3.line 124: in figure 3 c, the cracks are not very clear, like in figures 3b and d.Please improve the quality of the image.

4.line 127. Also in Figure 4c, it appears at a different magnitude than the other 3 images. Revise the figure.

  1. TEM images in Figure 13 could be presented with a revised scale bar.

The rest is fine!

Reviewer 2 Report

The comments/suggestions are as follows.

1.     In the abstract, at least one line of application and need should be added. Further, 4th sentence is very big and not easy to understand the meaning. It should be short and easy to understand. Moreover, the abstract should be more quantitative regarding the improvement observed.

2.     The author should mention the name of the etching agent in materials and methods and their quantity. Further, it needs to include data on repeatability. The author should mention the same for fatigue tests.

3.     The heading should be Results and Discussion, not only Results. It shows that the author has only discussed the obtained results.

4.     What does it mean of dash line circles? Mention it.

5.     The author should mention the value of lower surface roughness. Better mention the surface roughness for all thicknesses.

6.     In Fig 3 and Fig 4, the author mentions their observation inside the SEM images.

7.     A discussion of obtained Results is required. The author should add it for all the obtained results.

Reviewer 3 Report

The current work illustrates the effect of Ti layer thickness on the mechanical properties mainly (fatigue strength) of Al-based alloys. The authors tested many layers thickness from 1 to 5 µm. The authors illustrated that by increasing the Ti layer thickness an enhancement in mechanical properties is reported, where the fatigue life and fatigue limit increase proportionally to the thickness of the titanium coating due to changes in surface roughness and adhesion capability. The authors only concentrate on the morphological investigations of the samples. The manuscript with the present form is hardly to accept in Coating, many enhancements in over all the manuscript is needed as follow:

The introduction strongly needs to be improved with more information about the progress in enhancement of mechanical properties of Al-Si based alloys and different techniques and methods using to achieve this issue, then focus on the current method and using Ti coating layer in enhancement the mechanical properties and the fatigue life behavior. I hardly seen the novelty of using Ti coating layer to enhance the mechanical properties. In addition, the author should add the expecting applications of the high fatigue life for Ti coating layer with 5 µm.

The authors only focused on the morphological investigation by using SEM and TEM. The presentation of data is too confusing. I recommend the author reorganize the presentation of these data with clearer way to be easily understood for the readers. In addition, I recommend the authors calculate and estimate the average value of microstructure characteristics (Grain size/μm, SDAS/μm, …..) of alloys at different locations for all of the samples. In addition, TEM analysis need to be enhanced with more details and calculation of all microstructure parameters and plotted in table with Ti layer thickness parameter.

If possible the authors add XRD analysis for all of sample to match with SEM and TEM analysis,.

The authors should explain in detail the Fatigue test procedure and include the experimental data in the revised version of the manuscript.

The authors mentioned in the abstract that the increasing of Ti layer thickness change the surface roughness and adhesion capability without providing any experimental data support their point of view, it will be great if the authors add some experimental data by using AFM or surface roughness profile meter for all samples to see the changing of the magnitude of surface energy by increasing the Ti layer thickness. 

Round 2

Reviewer 2 Report

No Comments.

Reviewer 3 Report

The authors fixed all my comments correctly. Thus the manuscript can publish in the present form.